# Estimating the effectiveness of an enhanced 'Improving Access to Psychological Therapies' (IAPT) service addressing the wider determinants of mental health: a real-world evaluation

Alice Porter ,[1] Matthew Franklin ,[2] Frank De Vocht,[1,3] Katrina d'Apice,[1] Esther Curtin,[4] Patricia Albers,[1] Judi Kidger [1]

¹Population Health Sciences, University of Bristol, Bristol, UK
²Health Economic and Decision Science, The University of Sheffield, Sheffield, UK
³NIHR Applied Research Collaboration West (NIHR ARC West), Bristol, UK
⁴Population Health, Faculty of Epidemiology and Population Health, London School of Hygiene & Tropical Medicine, London, UK

**Correspondence to**
Dr Alice Porter;
alice.porter@bristol.ac.uk

## ABSTRACT

**Background** Addressing the wider determinants of mental health alongside psychological therapy could improve mental health service outcomes and population mental health.

**Objectives** To estimate the effectiveness of an enhanced 'Improving Access to Psychological Therapies' (IAPT) mental health service compared with traditional IAPT in England. Alongside traditional therapy treatment, the enhanced service included well-being support and community service links.

**Design** A real-world evaluation using IAPT's electronic health records.

**Setting** Three National Health Service IAPT services in England.

**Participants** Data from 17 642 service users classified as having a case of depression and/or anxiety at baseline.

**Intervention** We compared the enhanced IAPT service (intervention) to an IAPT service in a different region providing traditional treatment only (geographical control), and the IAPT service with traditional treatment before additional support was introduced (historical control).

**Primary outcome measures** Patient Health Questionnaire-9 (PHQ-9) Depression Scale (score range: 0–27) and Generalised Anxiety Disorder-7 (GAD-7) Anxiety Scale (score range: 0–21); for both, lower scores indicate better mental health. Propensity scores were used to estimate inverse probability of treatment weights, subsequently used in mixed effects regression models.

**Results** Small improvements (mean, 95% CI) were observed for PHQ-9 (depression) (−0.21 to −0.32 to −0.09) and GAD-7 (anxiety) (−0.23 to −0.34 to −0.13) scores in the intervention group compared with the historical control. There was little evidence of statistically significant differences between intervention control and geographical control.

**Conclusions** Embedding additional health and well-being (H&W) support into standard IAPT services may lead to improved mental health outcomes. However, the lack of improved outcomes compared with the geographical control may instead reflect a more general improvement to the intervention IAPT service. It is not clear from our findings whether an IAPT service with

## STRENGTHS AND LIMITATIONS OF THIS STUDY

⇒ This study is the first to estimate the effectiveness of embedding health and well-being support into the traditional National Health Service (NHS) mental health service in England.
⇒ This study used real-world electronic health record (EHR) data to explore the effectiveness of an enhanced Improving Access to Talking Therapies (IAPT) service.
⇒ Due to the nature of the research, this study was non-randomised and involved comparing the enhanced IAPT service (intervention) to the existing service (historical control), whereby potential changes to the service over time could not be accounted for.
⇒ Outcome measures were guided by IAPT's key performance indicators and EHRs. However, no data was available on the wider determinants of mental health, such as well-being and quality of life.

additional H&W support is clinically superior to traditional IAPT models.

## INTRODUCTION

Approximately one in six adults in England have a diagnosable mental health disorder (MHD), for which the prevalence and severity has worsened over the last 30 years, exacerbated by the COVID-19 pandemic.[1 2] Preventing and treating MHDs effectively is a global public health priority.

Depression and anxiety are globally considered common MHDs, due to their prevalence.[3] Evidence-based effective treatment for common MHDs includes psychological therapies, for example, cognitive–behavioural therapy or psychotherapy.[4] The National Health Service (NHS) England's *Improving Access to Psychological Therapies* (IAPT) programme provides evidence-based

psychological therapies for MHDs. IAPT treatment starts with a General Practitioner (GP) referral or self-referral, with subsequent initial assessment by IAPT staff for anxiety and/or depression 'caseness', followed by allocation to the waiting-list before the first treatment session can be provided. Those requiring IAPT treatment are initially offered a multi-session course with a psychological well-being practitioner (PWP), dependent on their symptom severity.[5]

IAPT defines anxiety and depression caseness as a service user reported score of ≥8 on the Generalised Anxiety Disorder-7 Questionnaire (GAD-7) and ≥10 on the Patient Health Questionnaire (PHQ-9), respectively. IAPT's patient-reported key performance indicators (KPIs) are based on the service users' first (ie, initial assessment) and last (ie, at discharge) PHQ-9 and GAD-7 scores; this includes:

► *Recovery*: moving from 'caseness' (PHQ-9≥10; GAD-7≥8) on either measure to 'no caseness' (PHQ-9<10; GAD-7<8) on both measures.
► *Reliable change*: decrease of ≥6 on PHQ-9 or ≥4 on GAD-7, without a score increase of PHQ-9 ≥6 or GAD-7 ≥4 on the other measure.
► *Reliable recovery*: achieving both recovery and reliable change.

Of 1.81 million adults referred to IAPT in 2021/2022, approximately 50% were defined as 'recovered' at the point of service discharge.[5] However, a large proportion of people do not show reliable change or recover by the point of IAPT discharge. Additionally, common MHD prevalence has not reduced since the IAPT programme started in 2008.[6]

Evidence suggests factors such as unemployment, poverty, debt, abuse, social isolation, poor health and well-being (H&W) and physical inactivity can cause and/or worsen mental health conditions[3] and therefore increase the need for services such as IAPT. The WHO suggests mental health services should consider these psychosocial factors when providing treatment.[3] To address this, a 'whole-person' approach to mental health is required, including strengthening the links between IAPT and wider community and well-being support.[7 8] Focusing on these wider determinants could help to improve the likelihood of recovery from mental health issues by addressing external factors causing these issues. A randomised controlled trial (RCT) found social prescribing (ie, making referrals from a primary healthcare service to a voluntary community service) reduced anxiety and improved quality-of-life, but cost more and did not reduce depression severity.[9] An evaluation of IAPT linked with employment services reported that service users who accessed employment support had better employment-related outcomes, with some indication of mental health improvements.[10]

As part of the mixed methods 'Assessing a Distinct Improving Access to Psychological Therapies service' study, we aimed to evaluate a new, locally enhanced IAPT service model, which introduced an additional H&W pathway to address wider determinants of mental health. The aim of the quantitative evaluation reported in this article was to estimate and report the effectiveness of the enhanced IAPT service compared with IAPT treatment as usual (TAU). We hypothesise that the enhanced IAPT service will lead to improved mental health outcomes and waiting times, due to its ability to address the wider determinants of mental health, as well as provide treatment for MHDs.

## METHODS

### Intervention

The enhanced IAPT service (intervention) was introduced in March 2021 in three local authorities in South West (SW) England and replaced the existing IAPT service. The enhanced IAPT service provides a more tailored and holistic treatment plan, whereby after an initial assessment with a PWP at baseline (ie, before waiting-list allocation), service users can be referred to IAPT TAU only (eg, psychotherapy), IAPT TAU plus H&W pathway or the H&W pathway alone. During the initial assessment, PWPs will seek to understand any external causes for a service users' anxiety or depression. For example, a service user presenting with anxiety due to large amounts of debt may be offered therapy to address their anxiety and be referred to debt management via the H&W pathway, whereas a service user with a more general form of anxiety may only be offered therapy. Therefore, the intervention group is made up of service users receiving any of these three treatment options within the enhanced IAPT service in SW England.

The H&W pathway consisted of two elements. First, the 'healthy living healthy minds' programme, a six-session group webinar series offering guided exercise and advice on healthy lifestyles. Second, and/or, one-to-one sessions with a 'well-being navigator', who facilitated access to community organisations to address the wider psychosocial problems individuals were experiencing (eg, poverty, unemployment and social isolation). Online supplemental figure 1 illustrates the enhanced IAPT service pathways. Further details of the H&W pathway are described in the qualitative evaluation.[11]

### Study design and target trial (TT) protocol

This was a service-led public health evaluation. A non-randomised study[12 13] was conducted using IAPT's electronic health records (EHRs). Treatment allocation (at service level) was non-randomised because it was not possible for the researchers to control its allocation, as the introduction of the enhanced IAPT service resulted from a new service provider and service redesign rather than a research study design (eg, RCT). The enhanced IAPT service (intervention) was compared with the previous IAPT service with TAU (historical control) and with a standard IAPT service with TAU in South-East England (geographical control). The historical control was provided by a different service provider and the

**Table 1** Target trial protocol summary

| Component | Description |
|---|---|
| Eligibility criteria | New referrals to IAPT: no attendance at the IAPT site in the previous 6 months since the new referral. Newly referred during: March 2021 to March 2022 (intervention & geographical control) or March 2018 to March 2019 (historical control). Condition caseness at baseline: service users classified as having depression caseness (PHQ-9≥10) or anxiety caseness (ie, GAD-7≥8) at baseline (ie, before waiting-list allocation). Baseline data: recorded PHQ-9 (depression severity) and GAD-7 (anxiety severity) score at baseline—see 'Condition caseness at baseline'. As-started treatment: service users had attended at least one treatment session to be defined as 'as-started' treatment. |
| Treatment strategies | Intervention: enhanced IAPT service (South West, England), as TAU plus 'healthy living healthy minds' programme and/or 1:1 well-being navigator sessions. Geographical control: TAU IAPT service in South East, England. Historical control: TAU IAPT service in the intervention area but before the enhanced service had been implemented. |
| Assignment procedures | Non-randomised and unblinded: referrals are assessed for eligibility by the service, if deemed eligible are offered TAU (across all groups) or TAU plus 'healthy living healthy minds' programme and/or 1:1 well-being navigator sessions (intervention only), with uptake based on service user preference. |
| Follow-up period(s) | Starts at baseline appointment to assess condition caseness and allocate people to the waiting-list before first treatment session, and ends at discharge from service or self-discharge from service. |
| Outcome(s) | Primary: PHQ-9 Score (continuous), GAD-7 Score (continuous) and WSAS Score (continuous). Secondary: reliable change (binary), recovery (binary variables) and waiting times (continuous variable). |
| Estimand(s) (casual contrasts) | As-started primary: in new referrals to IAPT, the between-group difference in mean PHQ-9 or GAD-7 or WSAS for those referred to the enhanced IAPT service compared with IAPT TAU in the geographical or historical control at the point of service discharge, regardless of TAU received. Sensitivity analyses: in new referrals to IAPT, what is the between-group difference in mean PHQ-9 or GAD-7 or WSAS for those who have been within the service for at least 12 or 16 or 20 or 24 weeks, referred to the enhanced IAPT service compared with IAPT TAU in the geographical or historical control at the point of service discharge, regardless of TAU received. |
| Analysis plan | Overall: an analytical plan was prespecified. Propensity scores: logistic regression estimated propensity scores were used to derive IPTWs. Weighted regressions: linear mixed effects weighted regression analyses were conducted for primary outcomes, with the weights based on IPTWs. Linear and logistic regression analyses were conducted for secondary outcomes. Unadjusted, fully adjusted and doubly robust (fully adjusted and weighted) model results were compared for each outcome. Subgroups: those within the intervention group who accessed: (1) the IAPT TAU only and (2) IAPT TAU plus the enhanced service (N.B. no service users were offered the enhanced service without TAU). Health inequalities: the primary outcome models were rerun with additional interactions between treatment group and age, gender, ethnicity or IMD. |

GAD-7, Generalised Anxiety Disorder questionnaire-7; IAPT, Improving Access to Psychological Therapies; IMD, Index of Multiple Deprivation; IPTWs, inverse probability of treatment weights; N.B., nota bene (ie, please note); PHQ-9, Patient Health Questionnaire-9; TAU, treatment as usual; WSAS, Work and Social Adjustment Scale.

geographical control by the same service provider as the intervention.

Table 1 provides an overview of the a priori specified TT framework protocol, which aims to reduce bias by clearly articulating seven study dimensions: eligibility criteria, treatment strategies, assignment procedure, follow-up period, outcomes, estimand(s) and analysis plan.[13] The TT framework is intended to mimic the ideal 'pragmatic' RCT. The study's planning, conducting and reporting follows the National Institute for Health and Care Excellence (NICE) real-world evidence (RWE) framework.[14] The reporting of studies conducted using observational rountinely collected health data statement for pharmaco-epidemiology (RECORD-PE) checklist is provided in the online supplemental table 1.[15]

### Measures

We used IAPT's EHR data, anonymised before sharing (ie, on IAPT premises) in line with information governance processes for secondary data analysis. Demographics (age, sex, ethnicity, postcode and employment status) were provided as baseline variables. Postcodes were matched to a national Index of Multiple Deprivation (IMD) dataset, as an indicator of socioeconomic status.[16]

Primary outcomes included service user reported scores as continuous variables:

► PHQ-9 Depression Score from 0 (best state) to 27 (worst state).
► GAD-7 anxiety scores from 0 (best state) to 21 (worst state).
► Work and Social Adjustment Scale (WSAS) from 0 (best state) to 40 (worst state).

Secondary outcomes included KPIs reliable change (binary), recovery (binary) and waiting time (continuous). *Reliable change* and *recovery* (based on PHQ-9 and GAD-7 scores) were calculated and reported only for those who reached the point of service discharge, in line with the NHS England's IAPT manual.[4] Waiting time was defined as the duration in days from referral to first appointment, as reported in the EHRs.

### Eligible participants
Eligible participants were referrals to IAPT classified as having depression (PHQ-9) and/or anxiety (GAD-7) 'caseness' at baseline, with 'baseline' defined as the initial assessment before waiting-list allocation, as part of TAU. For analyses focused on WSAS, eligible participants must have completed WSAS at baseline.

Intervention and geographical control participants were service users in the relevant areas who were referred between March 2021 and March 2022. Historical control participants were service users in the intervention area, referred between March 2018 and March 2019 (ie, before the enhanced service was introduced when the service was just IAPT TAU). Data from service users referred between April 2019 and February 2021 were not requested, to minimise spill-over effects from the historical control into the intervention group. All eligible service users had to have completed at least one treatment session to be included in the analyses.

### Statistical analyses
Descriptive and statistical analysis was conducted using Stata V.15. Data preparation included recoding and deriving variables, and merging datasets required for analysis.

Propensity scores were estimated using logistic regression with treatment assignment as the dependent variable and clinical (baseline PHQ-9, GAD-7 and WSAS) and demographic variables (age, sex, IMD and ethnicity) as independent variables. The propensity scores were used to calculate inverse probability of treatment weights (IPTWs) used within mixed effects regression models as a 'doubly robust' (ie, weighted regression) approach for balancing the comparison groups with respect to baseline covariates. This approach is defined as 'doubly robust' because combining the IPTWs and regression methods means only the weights or the regression need be correctly specified to obtain an unbiased effect estimator; thus minimising between-group baseline imbalances while accounting for baseline confounding due to the non-randomised treatment allocation.[17] Employment status was not included in the regression models due to the proportion of missing data (19%). Data were visually inspected to check the level of balance between groups after reweighting (see online supplemental figure 2).

Linear mixed effects weighted regression analyses were conducted for the primary outcomes, to account for repeated measures, with service user ID as the random effects variable. For the primary outcomes, a negative average treatment effect on the treated (ATT) represents an improvement to mental health. Linear and logistic weighted regressions were conducted for secondary outcomes. Model residuals were inspected to test assumptions of distribution, linearity and homoscedasticity. A complete-case analysis was conducted by focusing the analysis on people with at least one follow-up data collection timepoint, thus assuming missing outcome scores were missing completely at random. The follow-up period started at baseline and ended at discharge from the service, to align with IAPT's KPIs.[4]

### Sensitivity and subgroup analyses
Sensitivity analyses were conducted to explore outcomes among samples of service users who had been within the service for at least 12, 16, 20 and 24 weeks, respectively, with the follow-up period ending at discharge from the service. These analyses were based on evidence suggesting at least 8–12 weeks of treatment is required to observe clinical change in MHDs.[18–20]

Subgroup analyses included those within the intervention group who accessed: (1) the IAPT TAU only and (2) IAPT TAU plus H&W pathway (NB: no service users were offered the H&W pathway without TAU).

To explore potential health inequalities, the primary outcome models were re-run with additional interactions between treatment group and age, sex, ethnicity or IMD. Effect sizes, 95% CIs and p values were reported.

### Patient and public involvement
The development of the research proposal was overseen by a local IAPT steering group. The service provider and service users were involved in dissemination plans of the research.

## RESULTS
### Descriptive statistics
Table 2 presents demographic characteristics of the eligible participants, which were similar across intervention and control groups. Online supplemental table 2 presents the number of service users with data available in the IAPT EHRs, who met study eligibility criteria and who were included in analyses.

Table 2 also presents the mental health outcomes descriptively, showing that the mean baseline mental health scores were 1–2 points lower in the intervention versus control groups. The percentage of service users achieving PHQ-9 reliable change were similar for the intervention group and historical control (both 43%), but

**Table 2** Descriptive statistics for demographics and mental health measures scores for eligible sample

| Measure | Submeasure | Intervention | | IAPT TAU+H&W | | IAPT TAU only | | Geographical | | Historical | |
|---|---|---|---|---|---|---|---|---|---|---|---|
| | | N | % | N | % | N | % | N | % | N | % |
| N, overall | N/A | 8020 | 100 | 728 | 100 | 7292 | 100 | 3548 | 100 | 6074 | 100 |
| Sex | Female | 5558 | 69 | 494 | 68 | 5064 | 69 | 2461 | 69 | 3948 | 65 |
| Ethnicity | Asian | 223 | 3 | 30 | 4 | 193 | 3 | 105 | 3 | 183 | 3 |
| | Black | 175 | 2 | 29 | 4 | 146 | 2 | 121 | 3 | 175 | 3 |
| | Mixed | 271 | 3 | 26 | 4 | 245 | 3 | 88 | 2 | 192 | 3 |
| | Other | 100 | 1 | 13 | 2 | 87 | 1 | 23 | 1 | 62 | 1 |
| | White | 7109 | 89 | 610 | 84 | 6499 | 89 | 3132 | 88 | 5336 | 88 |
| Employment | Employed | 4633 | 58 | 382 | 52 | 4251 | 58 | 1812 | 51 | 3537 | 58 |
| | Unemployed | 923 | 12 | 163 | 22 | 760 | 10 | 758 | 21 | 1156 | 19 |
| | Retired | 261 | 3 | 37 | 5 | 224 | 3 | 235 | 7 | 291 | 5 |
| | Student | 679 | 8 | 35 | 5 | 644 | 9 | 141 | 4 | 397 | 7 |
| PHQ-9 RC* | No | 3346 | 57 | 414 | 62 | 2932 | 57 | 1467 | 50 | 3096 | 58 |
| | Yes | 2507 | 43 | 256 | 38 | 2251 | 43 | 1449 | 50 | 2276 | 42 |
| GAD-7 RC* | No | 2549 | 44 | 350 | 52 | 2199 | 42 | 1226 | 42 | 2594 | 48 |
| | Yes | 3304 | 56 | 320 | 48 | 2984 | 58 | 1690 | 58 | 2778 | 52 |
| Recovered* | No | 3019 | 52 | 485 | 72 | 2534 | 49 | 1507 | 52 | 3342 | 62 |
| | Yes | 2834 | 48 | 185 | 28 | 2649 | 51 | 1409 | 48 | 2030 | 38 |

| Measure | Submeasure | Intervention | | | IAPT TAU+H&W | | | IAPT TAU only | | | Geographical | | | Historical | | |
|---|---|---|---|---|---|---|---|---|---|---|---|---|---|---|---|---|
| | | Mean | SD | Range | Mean | SD | Range | Mean | SD | Range | Mean | SD | Range | Mean | SD | Range |
| Age | N/A | 35 | 14 | 15–92 | 40 | 15 | 16–89 | 35 | 13 | 15–92 | 39 | 16 | 15–107 | 37 | 14 | 16–93 |
| IMD | Decile | 5.5 | 1.9 | – | 5.3 | 1.9 | – | 5.5 | 1.9 | – | 5.7 | 2.3 | – | 5.2 | 1.8 | 2.4–9.1 |
| PHQ-9 | Baseline | 15 | 5 | 0–27 | 17 | 5 | 1–27 | 14 | 5 | 0–27 | 16 | 5 | 0–27 | 16 | 5 | 0–27 |
| | Last obs. | 11 | 6 | 0–27 | 13 | 7 | 0–27 | 10 | 6 | 0–27 | 11 | 7 | 0–27 | 11 | 7 | 0–27 |
| | Dif. | –4.1 | 5.7 | –26 to 22 | –4.0 | 5.9 | –25 to 18 | –4.1 | 5.7 | –26 to 22 | –5.2 | 6.3 | –27 to 15 | –4.4 | 6 | –26 to 20 |
| GAD-7 | Baseline | 13 | 4 | 0–21 | 14 | 5 | 0–21 | 13 | 4 | 0–21 | 14 | 4 | 0–21 | 14 | 4 | 0–21 |
| | Last obs. | 9 | 6 | 0–21 | 11 | 6 | 0–21 | 9 | 6 | 0–21 | 10 | 6 | 0–21 | 10 | 6 | 0–21 |
| | Dif. | –3.9 | 5.4 | –21 to 17 | –3.4 | 5.3 | –21 to 14 | –4.0 | 5.4 | –21 to 17 | –4.7 | 5.8 | –21 to 14 | –3.9 | 5.7 | –21 to 19 |
| WSAS | Baseline | 18 | 8 | 0–40 | 21 | 9 | 0–40 | 18 | 8 | 0–40 | 19 | 9 | 0–40 | 20 | 9 | 0–40 |
| | Last obs. | 15 | 9 | 0–40 | 18 | 10 | 0–40 | 14 | 9 | 0–40 | 15 | 10 | 0–40 | 16 | 10 | 0–40 |
| | Dif. | –3.2 | 8.4 | –38 to 37 | –3.0 | 9.1 | –28 to 30 | –3.2 | 8.3 | –38 to 37 | –4.3 | 9.3 | –40 to 36 | –4.4 | 9.6 | –39 to 40 |
| Waiting time (days) | To first app. | 20 | 25 | 0–336 | 19 | 21 | 0–240 | 20 | 26 | 0–336 | 10 | 20 | 0–322 | 18 | 20 | 0–216 |

Continued

**Table 2** Continued

| Measure | Submeasure | Mean | SD | Range | Mean | SD | Range | Mean | SD | Range | Mean | SD | Range | Mean | SD | Range | Mean | SD | Range |
|---|---|---|---|---|---|---|---|---|---|---|---|---|---|---|---|---|---|---|---|
| Sessions attended, To discharge n | | 5 | 3 | 1–32 | 5 | 3 | 1–32 | 8 | 6 | 1–32 | 5 | 3 | 1–27 | 5 | 3 | 1–25 | 6 | 4 | 1–30 |

*Includes patients who have been discharged from the service only.

GAD-7, Generalised Anxiety Disorder Questionnaire 7; H&W, Health & Wellbeing pathway; IAPT, Improving Access to Psychological Therapies; IMD, Index of Multiple Deprivation; PHQ-9, Physical Health Questionnaire-Depression Scale; RC, reliable change (defined as the reduction of six points or more on the PHQ-9 or four points or more on the GAD-7); TAU, treatment as usual; WSAS, Work and Social Adjustment Scale. Recovered, defined as moving below the threshold of 10 on PHQ-9 or 8 on GAD-7.

higher for the geographical control (50%). Percentage of service users achieving GAD-7 reliable change: intervention, 56%; geographical, 58%; historical, 52%. The percentage who recovered was similar for the intervention group and geographical control (both 48%), but lower for historical control (38%).

### Primary analyses

Table 3 indicates there was little difference in the ATT for PHQ-9, GAD-7 and WSAS when comparing the intervention group to geographical control. Compared with the historical control, we observed a small, negative ATT for PHQ-9 (−0.40, 95% CI −0.53 to −0.27), GAD-7 (−0.43, 95% CI −0.55 to −0.32) and WSAS (−0.53, 95% CI −0.53 to −0.15), representing small improvements in mental health in the intervention group.

Table 3 indicates that the odds of achieving PHQ-9 or GAD-7 reliable change was similar between intervention group and geographical control. The odds of recovery was lower in the intervention group versus geographical control (OR=0.87, 95% CI 0.78 to 0.96). Conversely, the odds was higher in the intervention group versus historical control for achieving PHQ-9 reliable change (OR=1.22, 95% CI 1.12 to 1.33), GAD-7 reliable change (OR=1.17, 95% CI 1.17 to 1.38) and recovery (OR=1.46, 95% CI 1.34 to 1.59). Waiting-times were longer in the intervention group versus geographical control (mean=10 days, 95% CI 9.21 to 11.52) and marginally longer versus historical control (mean=1.5 days, 95% CI 0.67 to 2.41).

### Sensitivity analyses

Table 4 presents the sensitivity analysis results, which indicate no difference in ATT between intervention group and geographical control, except for a small negative ATT (ie, improved score) for GAD-7 in the 24-week sample (−0.22, 95% CI −0.39 to −0.05). Comparing the intervention group to historical control, small negative ATTs (ie, improved scores) were observed for PHQ-9, GAD-7 and WSAS across all four samples, although associations were weak for WSAS in the 16-week to 24-week samples.

The odds of achieving PHQ-9 reliable change was higher in the intervention group versus geographical control in the 24 week sample only (OR=1.15, 95% CI 1.02 to 1.30). The odds of achieving GAD-7 reliable change was higher in the intervention group versus geographical control in the 20-week (OR=1.17, 95% CI 1.04 to 1.32) and 24-week samples (OR=1.20, 95% CI 1.05 to 1.36). The odds of recovery was not different between intervention group and geographical control. The odds of achieving PHQ-9 reliable change, GAD-7 reliable change and recovery was higher in the intervention group versus historical control across all four samples.

Across the four samples, findings suggest 10–12 days longer waiting-times in the intervention group compared with the geographical control. Waiting times were slightly longer by 1.6–3.0 days in the intervention group compared with historical control in the 16-week to 24-week samples, but no difference was observed in the 12-week sample.

**Table 3** Doubly robust mixed effects linear regression models for primary and secondary outcomes in as-started sample

| | Intervention versus geographical control | | | | Intervention versus historical control | | | |
| --- | --- | --- | --- | --- | --- | --- | --- | --- |
| | n, obs=48 652; n, patients=8542 | | | | n, obs=64 044; n, patients=10 951 | | | |
| | Coefficient | 95% CI | | P value | Coefficient | 95% CI | | P value |
| PHQ-9 | −0.08 | −0.23 | 0.07 | 0.284 | −0.40 | −0.53 | −0.27 | <0.001 |
| GAD-7 | −0.13 | −0.27 | 0.00 | 0.052 | −0.43 | −0.55 | −0.32 | <0.001 |
| WSAS | 0.09 | −0.14 | 0.32 | 0.437 | −0.34 | −0.53 | −0.15 | <0.001 |
| PHQ-9 reliable change | 0.90 | 0.82 | 1.00 | 0.045 | 1.22 | 1.12 | 1.33 | <0.001 |
| GAD-7 reliable change | 1.00 | 0.90 | 1.10 | 0.941 | 1.27 | 1.17 | 1.38 | <0.001 |
| Recovery | 0.87 | 0.78 | 0.96 | 0.007 | 1.46 | 1.34 | 1.59 | <0.001 |
| Waiting times | 10.37 | 9.21 | 11.52 | <0.001 | 1.54 | 0.67 | 2.41 | 0.001 |

All models are adjusted for covariates (age, gender, ethnicity, Index of Multiple Deprivation, baseline PHQ-9, baseline GAD-7 and baseline WSAS). Inverse propensity score weighting applied to all models, weighted for all covariates.
Sample includes all patients who met the study inclusion criteria, with follow-up as the point of service discharge.
Reliable change PHQ-9 is defined as a reduction of ≥6 points. For patients to have a measure of reliable change PHQ-9, they must have been discharged.
Reliable change GAD-7 is defined as a reduction of ≥4 points. For patients to have a measure of reliable change GAD-7, they must have been discharged.
Recovery is defined as entering the service at the threshold of a score of ≥10 on PHQ-9 and/or ≥8 on GAD-7 and moving to 'no-caseness', being a score below the threshold on both measures of PHQ-9 and GAD-7. For a patient to have a measure of recovery, they must have been discharged.
GAD-7, Generalised Anxiety Disorder-7; PHQ-9, Patient Health Questionnaire-9; WSAS, Work and Social Adjustment Scale.

## Subgroup analyses and investigating health inequalities

Table 2 shows that within the intervention group, the percentage of service users achieving PHQ-9 or GAD-7 reliable change was lower among those who received IAPT TAU plus H&W pathway versus IAPT TAU only. However, table 2 also shows that those who received IAPT TAU plus H&W pathway had slightly higher mean PHQ-9 and GAD-7 scores at baseline.

Online supplemental table 3 shows a group by gender interaction was observed for PHQ-9 and GAD-7 when comparing the intervention group to geographical control, suggesting a slightly greater ATT among males in the geographical control. A group by IMD interaction was observed for GAD-7 and WSAS when comparing intervention group to historical control, suggesting a slightly greater ATT among service users living in areas of low deprivation in the intervention group.

## DISCUSSION

Using a non-randomised study design and routinely collected 'real-world' IAPT EHR data, this study is the first to evaluate the effectiveness of providing an array of support to address the wider determinants of mental health in addition to psychological therapies. When comparing the intervention to the historical control, our findings indicate that the enhanced service led to greater (although small) improvements in mental health scores and a higher number of service users recovering from their mental disorder. This aligns with the findings from the qualitative evaluation, which suggests the H&W pathway was perceived by service deliverers and users to have a positive impact on mental health.[11] However, when

comparing the intervention to the geographical control over the same time period, we observed little evidence of improvement.

Previous studies have shown IAPT service adaptation can address harmful health behaviours (eg, smoking)[21] and help treat other mental health-related issues (eg, insomnia).[22] Our evaluation[11] provides evidence that IAPT services can be adapted to provide person centred, tailored links to services, to address the underlying reasons for poor mental health, which meets key policy[7] and WHO recommendations.[3] However, in this study the evidence for an additional benefit over psychological therapy is limited, as we observed small improvements in the historical control only. As the historical control service was taken over by a new service provider (the same service provider as the geographical control), it may be that the standard elements of the intervention service, such as the psychological therapy, had been improved by the new service provider, which could explain why we observe mental health improvements in the intervention compared with the historical control but not the geographical control. In other words, we cannot rule out that the effects seen were due to an improved standard service, rather than the enhanced aspects, especially as recovery rates in the historical control service sample were lower than the national average (38% vs 53%)[5]

The reliable change and recovery results were more pronounced than the ATTs from the continuous outcome measure scores. The odds of achieving reliable change was between 11% and 27% and achieving recovery was between 30% and 47% greater in the intervention group versus historical control analyses, while the differences

**Table 4** Doubly robust mixed effects linear regression models for primary and secondary outcomes for patients within the service for at least 12–24 weeks

| Measure | 12 weeks | | | | 16 weeks | | | | 20 weeks | | | | 24 weeks | | | |
|---|---|---|---|---|---|---|---|---|---|---|---|---|---|---|---|---|
| | Coef. | 95% CI | | p value | Coef. | 95% CI | | p value | Coef. | 95% CI | | p value | Coef. | 95% CI | | p value |
| Intervention group versus geographical control | | | | | | | | | | | | | | | | |
| PHQ-9 | -0.11 | -0.29 | 0.07 | 0.229 | -0.13 | -0.31 | 0.04 | 0.131 | -0.10 | -0.28 | 0.08 | 0.268 | -0.19 | -0.38 | 0.01 | 0.056 |
| GAD-7 | -0.10 | -0.26 | 0.06 | 0.216 | -0.12 | -0.28 | 0.04 | 0.127 | -0.12 | -0.29 | 0.04 | 0.143 | -0.22 | -0.39 | -0.05 | 0.012 |
| WSAS | 0.20 | -0.07 | 0.46 | 0.141 | 0.03 | -0.24 | 0.29 | 0.833 | 0.00 | -0.27 | 0.28 | 0.978 | -0.10 | -0.39 | 0.18 | 0.476 |
| PHQ-9 RC | 1.05 | 0.93 | 1.18 | 0.459 | 1.06 | 0.94 | 1.19 | 0.351 | 1.12 | 0.99 | 1.26 | 0.062 | 1.15 | 1.02 | 1.30 | 0.027 |
| GAD-7 RC | 1.09 | 0.97 | 1.23 | 0.156 | 1.11 | 0.99 | 1.25 | 0.085 | 1.17 | 1.04 | 1.32 | 0.012 | 1.20 | 1.05 | 1.36 | 0.005 |
| Recovery | 1.07 | 0.94 | 1.21 | 0.288 | 1.03 | 0.91 | 1.16 | 0.670 | 1.13 | 1.00 | 1.28 | 0.053 | 1.11 | 0.98 | 1.26 | 0.113 |
| Waiting times | 10.04 | 9.03 | 11.05 | <0.001 | 10.88 | 9.94 | 11.81 | <0.001 | 11.23 | 10.36 | 12.10 | 0.000 | 11.06 | 10.19 | 11.94 | <0.001 |
| Intervention group versus historical control | | | | | | | | | | | | | | | | |
| PHQ-9 | -0.46 | -0.61 | -0.31 | <0.001 | -0.35 | -0.50 | -0.20 | <0.001 | -0.32 | -0.48 | -0.17 | <0.001 | -0.31 | -0.48 | -0.15 | <0.001 |
| GAD-7 | -0.41 | -0.55 | -0.27 | <0.001 | -0.35 | -0.48 | -0.21 | <0.001 | -0.30 | -0.44 | -0.16 | <0.001 | -0.32 | -0.46 | -0.17 | <0.001 |
| WSAS | -0.39 | -0.61 | -0.17 | 0.001 | -0.27 | -0.49 | -0.05 | 0.019 | -0.25 | -0.48 | -0.02 | 0.036 | -0.25 | -0.50 | -0.01 | 0.043 |
| PHQ-9 RC | 1.26 | 1.14 | 1.40 | <0.001 | 1.16 | 1.05 | 1.28 | 0.003 | 1.13 | 1.02 | 1.24 | 0.018 | 1.11 | 1.00 | 1.23 | 0.045 |
| GAD-7 RC | 1.21 | 1.10 | 1.34 | <0.001 | 1.19 | 1.08 | 1.31 | <0.001 | 1.11 | 1.01 | 1.23 | 0.034 | 1.14 | 1.03 | 1.27 | 0.013 |
| Recovery | 1.47 | 1.33 | 1.64 | <0.001 | 1.34 | 1.22 | 1.49 | <0.001 | 1.34 | 1.21 | 1.49 | <0.001 | 1.30 | 1.17 | 1.45 | <0.001 |
| Waiting times | 0.30 | -0.59 | 1.19 | 0.511 | 1.57 | 0.75 | 2.39 | <0.001 | 2.49 | 1.71 | 3.27 | <0.001 | 3.01 | 2.22 | 3.79 | <0.001 |

Intervention group versus geographical control, number of patient/number of obs: 12 weeks, 6021/35 999; 16 weeks, 6364/37 965; 20 weeks, 5972/36 310; 24 weeks, 5499/33 829.

Intervention group versus historical control, number of patient / number of obs: 12 weeks, 7805/47 512; 16 weeks, 8076/50 071; 20 weeks, 7603/47 835; 24 weeks, 6880/43 873.

All models are adjusted for covariates (age, gender, ethnicity, Index of Multiple Deprivation, baseline PHQ-9, baseline GAD-7 and baseline WSAS). Inverse propensity score weighting applied to all covariates, weighted for all covariates.

Samples include all patients who met the study inclusion criteria and were in the service for at least 12, 16, 20 and 24 weeks, respectively, with follow-up as the point of service discharge.

RC PHQ-9 is defined as a reduction of ≥6 points. For patients to have a measure of reliable change PHQ-9, they must have been discharged.

RC GAD-7 is defined as a reduction of ≥4 points. For patients to have a measure of reliable change GAD-7, they must have been discharged.

Recovery is defined as entering the service at the threshold of a score of ≥10 on PHQ-9 and/or ≥8 on GAD-7 and moving to a 'no-caseness', being a score below the threshold on both measures of PHQ-9 and GAD-7. For a patient to have a measure of recovery, they must have been discharged.

GAD-7, Generalised Anxiety Disorder-7; PHQ-9, Patient Health Questionnaire-9; RC, reliable change; WSAS, Work and Social Adjustment Scale.

in mean PHQ-9 and GAD-7 scores between groups were minimal. The ATTs we observed were smaller compared with a similar study, which evaluated the effectiveness of additional insomnia support alongside TAU within IAPT on clinical mental health outcomes.[22] Stott *et al*[22] reported ATTs of between −0.78 and −1.30 for PHQ-9, GAD-7 and WSAS (sample size of 1020) compared with our estimates, which were between −0.34 and −0.43. Our findings suggest a large individual variability in treatment effectiveness and may partly explain the lack of average improvement observed at the group level. The reasons for this variability likely go beyond the data collected by IAPT (such as demographics and number of sessions attended), and may include factors such as service user relationships with practitioners and linked services, and service users' cognition, attitudes and beliefs.[23] Our qualitative evaluation shed some light by highlighting certain implementation issues, such as service users and wider services expressing they lacked clarity and reasoning from PWPs about why a certain referral pathway was chosen.[11] These findings may help to explain the small ATTs and lack of statistical difference compared with the geographical control.

Previous evidence suggests that 8–12 weeks is required to observe clinically meaningful changes for common MHDs.[18–20] However, our sensitivity analyses suggest treatment duration may have to be longer. We observed small but favourable intervention effects for anxiety severity and reliable change compared with the geographical control among service users receiving treatment for at least 24 weeks. A longer duration may be required to properly establish service users' links into community services, and associated additional time from the point of addressing the wider determinants of people's mental health issues to the eventual impact on actual mental health outcome scores (eg, PHQ-9 and GAD-7 scores).

Our findings suggest further adaptation or additional resources may be required to address potential health inequalities. Specifically, we found a greater improvement in mental health among service users in the intervention compared with historical controls, but only for those from a higher socioeconomic background. Relatedly, a systematic review and meta-analysis suggested socioeconomic deprivation was associated with poorer treatment outcomes for mental health.[24] Future research should explore and address the barriers faced by service users living in deprived areas in achieving optimal treatment effects.

## STRENGTHS AND LIMITATIONS

An RCT was not possible as the enhanced IAPT service was part of a local service redesign, thus treatment allocation could not be controlled by the researchers. Instead, we used a quasi-experimental design based on real-world data (ie, IAPTs EHRs) to estimate the effectiveness of the enhanced IAPT service. This approach has causal inference implications due to the observational nature of the study,

which is more prone to bias (eg, study-entry selection and confounding bias) than RCTs. However, real-world studies have benefits including representing the effectiveness of the intervention under real-world scenarios and using real-world data, which helps improve the external validity (eg, generalisability) of the results within the setting of interest, for example, IAPT and NHS England. We used the TT framework[12] and NICE's RWE[14] framework to guide our study design and choice of analytical methods, including how we assessed and accounted for bias due to the non-randomised nature of the study. For example, we used a TT protocol, two control groups and 'doubly robust' weighted regression models, all of which are designed to reduce the potential impact of study-entry selection bias and confounding bias. Additionally, conducting prespecified sensitivity analyses with secondary outcomes and different samples increased our ability to assess result robustness. However, as is common, we used a 'no unmeasured confounders' assumption, thus the potential impact of unmeasured confounding has not been fully assessed.[25]

Two control groups were used; however, both have limitations to note: we cannot account for potential changes over time in the historical control and for potential clustering effects in the geographical control. Outcome score at service discharge was selected as the follow-up endpoint to align with IAPT's KPIs of interest, that is, based on PHQ-9 and GAD-7 scores at service discharge. This may have resulted in informative censoring because we do not know if and how service users' mental health may have changed after the point of discharge. In addition, service users may have been discharged for reasons other than recovery, which was not indicated in IAPT's EHR. For example, early discharge could be due to mental health improvement, or self-discharge due to service dissatisfaction, both impacting on the service users' outcome score at point of service discharge and beyond discharge. The use of more complex methods such as G-methods (eg, inverse probability of censoring weighting) required to account for informative censoring were beyond the scope of this study, given these methods have commonly been applied more to time-to-event analyses related to survival than patient-reported outcome measure scores. The application of such methods to outcome measures is an area for future research.

In terms of data suitability, the IAPT's EHR data were preferable compared with primary data collection as it allowed for a large sample size, quicker evaluation and minimal participant burden. However, dropping variables and associated participants with too much missing data (eg, employment status) reduced the study sample size and may have introduced some selection bias if our missing completely at random assumption does not hold. In addition, the EHR data did not include measures such as well-being, debt management or social isolation, which may have been useful to better understand the mechanistic impact of the H&W pathway.

Although the intervention group was not necessarily defined by those who received the H&W pathway (ie, also

included those who received therapy only), we should highlight that only 9.1% of the intervention group received H&W support. This may have been because PWPs did not deem patients to require the H&W service. However, it may have also been due to issues regarding intervention delivery. The enhanced service was newly introduced at the time of study, with the process of establishing links with wider community services still ongoing. This, in addition to the COVID-19 pandemic impacting service delivery, could have reduced the ability to link service users to other support.[26] This could have reduced estimates of effectiveness and provides further evidence that our results represent a more general improvement to the enhanced service versus historical service rather than a positive effect of the H&W support more specifically.

### Clinical implications

There is evidence supporting the role of social prescribing through GPs[9] for individuals with physical and mental comorbidities.[27] This study provides limited evidence that linking clinical mental health services, like IAPT, to public health community services addressing wider determinants of mental health may improve mental health outcomes. Stronger inferences cannot be made, as we observed improvements in mental health when comparing the intervention to the historical control but not to the geographical control. Longer term evaluation may be required to allow the enhanced IAPT service to be fully established for any impact to be clear. Mental health services which embed support to address the wider determinants should also consider including additional measures such as well-being and social isolation, in order to better understand the pathway to improved mental health.

### CONCLUSION

This study provides evidence of an improved IAPT service over time. However, evidence is less clear that embedding additional H&W support leads to improved service user mental health outcomes. Longer follow-up periods may be required to observe any further mental health improvements gained from H&W support in addition to psychological therapies.

**Acknowledgements** We would like to thank the IAPT service providers for providing anonymised electronic health record data and information about the data used in our analyses.

**Contributors** JK, FDV and MF were responsible for study conceptualisation and funding acquisition. AP, EC, Kd'A and PA were responsible for project administration. PA, Kd'A, FDV, MF, AP and JK contributed to methodology and the analysis plan. AP and MF conducted the analyses. AP wrote the first draft of the manuscript. JK is responsible for the overall content as the guarantor. All authors contributed to and approved the final manuscript.

**Funding** This study was supported by the National Institute for Health and Care Research (NIHR) School for Public Health Research (SPHR), Grant Reference Number SPHR-PHPES004-IAPT. The views expressed are those of the author(s) and not necessarily those of the NIHR or the Department of Health and Social Care.

**Competing interests** None declared.

**Patient and public involvement** Patients and/or the public were involved in the design, or conduct, or reporting, or dissemination plans of this research. Refer to the Methods section for further details.

**Patient consent for publication** Not applicable.

**Ethics approval** The Health Research Authority and Health and Care Research Wales granted ethical approval (reference number: 21/PR/0230).

**Provenance and peer review** Not commissioned; externally peer reviewed.

**Data availability statement** Data may be obtained from a third party and are not publicly available. All electronic health record data were provided by Improving Access to Psychological Therapies (IAPT) service providers. The terms of our data agreement with IAPT mean we cannot share these data but data may be requested from the service providers.

**ORCID iDs**
Alice Porter http://orcid.org/0000-0001-5281-7694
Matthew Franklin http://orcid.org/0000-0002-2774-9439
Judi Kidger http://orcid.org/0000-0002-1054-6758

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
