## [Reviewer comments · BMJ Open]

This paper was submitted to a another journal from BMJ but declined for publication following peer review. The authors addressed the reviewers' comments and submitted the revised paper to BMJ Open. The paper was subsequently accepted for publication at BMJ Open.

(This paper received two reviews from its previous journal and both reviewers agreed to published their review.)

ARTICLE DETAILS

TITLE (PROVISIONAL)	Estimating the effectiveness of an enhanced 'Improving Access to Psychological Therapies' (IAPT) service addressing the wider determinants of mental health: a real-world evaluation
AUTHORS	Porter, Alice; Franklin, Matthew; De Vocht, Frank; d'Apice, Katrina; Curtin, Esther; Albers, Patricia; Kidger, Judi

VERSION 1 – REVIEW

REVIEWER	Eiji Shimizu Chiba University
REVIEW RETURNED	04-Apr-2023

GENERAL COMMENTS	The authors investigated changes of depression and anxiety symptoms measured by PHQ-9 and GAD-7 in an enhanced 'Improving Access to Psychological Therapies' (IAPT) mental health service, compared to IAPT treatment-as-usual (TAU) in a different region (geographical-control), and the IAPT service with TAU before additional support (historical-control) using real world data. The following points should be criticized. 1) Regarding to the subgroup analyses, the authors described that the percentage of service-users achieving PHQ-9 or GAD-7 reliable change was lower among those who received IAPT TAU plus Health and Wellbeing (H&W) pathway vs IAPT TAU only in the intervention group. If possible, they should describe coefficient, 95% CI and p value of PHQ-9, GAD-7and WSAS in IAPT TAU plus Health and Wellbeing (H&W) pathway vs IAPT TAU only in a similar way in Table 3. 2) In Supplementary Figure 1, Psychological Wellbeing Practitioner (PWP) assessed whether H&W intervention is indicated or not. The authors should describe the decision criteria of H&W intervention in detail. 3) In Table 2, the authors showed us that there were 728 patients of IAPT TAU plus H&W pathway and 7292 patients of IAPT TAU only in the intervention group (total n =8020). That means approximately 9.1% of patients received IAPT TAU plus H&W pathway in the intervention group. The authors should explain and discuss whether
---

	the percentage is acceptable or not in detail. In an ideal manner, did 100% of patients receive IAPT TAU plus H&W pathway in the intervention group?
--	--

REVIEWER	Masaya Ito National Center of Neurology and Psychiatry, National Center for Cognitive-Behavior Therapy and Research
-----------------	--

REVIEW RETURNED	09-Apr-2023
-------------

GENERAL COMMENTS	Thank you for allowing me to review this significant manuscript entitled "Estimating the Effectiveness of an Enhanced 'Improving Access to Psychological Therapies (IAPT) Service Addressing the Wider Determinants of Mental Health: A Real-World Evaluation.'" This paper investigates the effectiveness of the enhanced interventions within the 'Healthy Living Healthy Minds' program and 'Wellbeing Navigator' sessions in comparison to the standard IAPT service. The study evaluates the enhanced service and standard service (i.e., IAPT Treatment-As-Usual) concerning depression, anxiety, and functioning outcomes, utilizing historical and geographical comparisons. The hypotheses tested bear substantial public and political importance due to the extensive human resources involved in the IAPT and enhanced IAPT service. While I recognize the researchers' mission and this study's primary aim, a scientific paper should offer more theoretical background and detailed information for readers to comprehend its scientific implications. My initial impression of this manuscript is the absence of specific hypotheses. Merely adding something may lead to better results, but the theoretical basis for this should be elucidated. Firstly, I question the selection of outcomes. The rationale for choosing depression and anxiety measures as outcomes remain unclear, as the enhanced care within the 'Healthy Living Healthy Minds' (HWHB) program and 'Wellbeing Navigator' (WBN) intervention do not seem to target depression and anxiety directly. If these are the intended outcomes, additional treatment should involve higher-intensity treatments for depression and anxiety. Is there any evidence supporting the efficacy of these interventions in improving depression and anxiety? If so, are the HWHB and WBN interventions superior to offering higher-intensity treatment (e.g., additional CBT)? It appears logical if the HWHB and WBN interventions target other aspects or determinants of mental health, such as well-being or quality of life, which the IAPT-TAU may not sufficiently improve. If so, the study's outcomes should focus on well-being and quality-of-life measures rather than depression and anxiety. The World Health Organisation's whole-person approach highlights the importance of examining various mental health or human living aspects. The HWHB and WBN interventions are indeed valuable; however, their importance is more proximal to participants' daily and social health status. Depression and anxiety outcomes seem distal for these additional interventions. At the very least, these discussions prove useful when discussing this study's results, implying that different outcome evaluations are necessary to assess the true outcomes of these enhanced services. The authors should estimate the effective strength of the enhanced service based on previous studies. Without this information, readers cannot evaluate whether the study has an adequate sample size or whether the benefits observed are "small" or not.
---

Details on the interventions are scant. I could not ascertain the nature of the six-session group webinar series within the 'Healthy Living Healthy Minds' program and one-to-one sessions of 'Wellbeing Navigator.' What were the contents? Who provided these services? How were these providers trained? What underpinning theories were used for these interventions? How was fidelity assessed for these interventions? How long did each session last? What types of psychosocial problems were addressed in the intervention group? What was the frequency of sessions? How long did the intervention last? Was there any interaction between participants in the 'Healthy Living Healthy Minds' program?

I acknowledge that reliable change and recovery have been used as IAPT outcomes. However, I require clarification on the differences between these two outcomes. Why were both outcomes necessary? In the discussion section, the distinction between mean PHQ-9 and GAD-7 scores and recovery status was helpful. I understand that the mean score represents the average improvement observed at the group level, while reliable change and recovery indicate the proportion of participants who responded to the intervention. Reliable change pertains to participants who responded to the intervention, whereas recovery pertains to participants who scored below the cut-offs. Is this understanding correct? If so, the proportion of recovery should always be lower than the reliable change. However, the results do not consistently support this in every group, possibly due to using both PHQ-9 and GAD-7 scores to define recovery status. The actual value of the proportion is not significantly different, leading me to question the importance of employing these two outcomes. Are there any specific hypotheses based on these different outcomes?

Please clarify why reliable change and recovery status was calculated only for those who reached the point of service discharge. This approach appears to be a completer analysis. It may be more appropriate to use an intention-to-treat (ITT) analysis incorporating all participants who started the service for statistical analyses.

I require an explanation of why waiting times could be considered an outcome. Please provide the rationale for hypothesizing that enhanced care is superior to TAU in this outcome and include an explanation.

Please include explanations of NB and IMD for Table 1.

In the introduction section, the authors mention that approximately 50% of the referred 1.81 million adults were defined as 'recovered' at the point of discharge, while the historical control group in this study achieved only 38% 'recovered' status. This discrepancy suggests that the enhanced service's superior results may be due to unusually weak results in the historical group. The authors should address this possibility. If this is likely, the significant results do not reflect the effectiveness of the enhanced service; rather, they may indicate the historical group's low-quality IAPT service provision. Although I understand that the researchers lack data on the quality and adherence of IAPT services specific to this study, are there any other findings regarding IAPT service quality that could help interpret the historical and geographical aspects?

VERSION 1 – AUTHOR RESPONSE

Comments from Reviewer 1

'The authors investigated changes of depression and anxiety symptoms measured by PHQ-9 and GAD-7 in an enhanced 'Improving Access to Psychological Therapies' (IAPT) mental health service, compared to IAPT treatment-as-usual (TAU) in a different region (geographical-control), and the IAPT service with TAU before additional support (historical-control) using real world data. The following points should be criticized.

1) Regarding to the subgroup analyses, the authors described that the percentage of service-users achieving PHQ-9 or GAD-7 reliable change was lower among those who received IAPT TAU plus Health and Wellbeing (H&W) pathway vs IAPT TAU only in the intervention group. If possible, they should describe coefficient, 95% CI and p value of PHQ-9, GAD-7 and WSAS in IAPT TAU plus Health and Wellbeing (H&W) pathway vs IAPT TAU only in a similar way in Table 3.

2) In Supplementary Figure 1, Psychological Wellbeing Practitioner (PWP) assessed whether H&W intervention is indicated or not. The authors should describe the decision criteria of H&W intervention in detail.

3) In Table 2, the authors showed us that there were 728 patients of IAPT TAU plus H&W pathway and 7292 patients of IAPT TAU only in the intervention group (total n =8020). That means approximately 9.1% of patients received IAPT TAU plus H&W pathway in the intervention group. The authors should explain and discuss whether the percentage is acceptable or not in detail. In an ideal manner, did 100% of patients receive IAPT TAU plus H&W pathway in the intervention group?'

Response to Reviewer 1 comments 1 & 3

We have combined our response to points 1 and 3 as they pertain to similar parts of the manuscript. To clarify, the intervention is not necessarily defined as receiving the Health & Wellbeing (H&W) pathway, rather the intervention represents a more tailored and holistic service than the standard IAPT service offered elsewhere, whereby the Psychological Wellbeing Practitioner (PWP) will decide based on a patients' initial assessment what course of treatment is likely to be most beneficial based on their mental health assessment but also the potential external factors associated with their mental health condition. This treatment can include therapy (as would be the case in all IAPT services) alone, or therapy alongside the H&W support or the H&W support alone. Therefore, the intervention group is made up of those patients who received, what is defined as this 'enhanced' (i.e. more tailored and holistic) service.

As you mention, 728 patients in the intervention group received therapy plus the H&W support and 7292 patients in the intervention group received therapy only (i.e. without H&W support). It is not necessarily the case that 100% of patients ideally received the H&W pathway because treatment plans are tailored based on individual need and not all patients referred to IAPT may have required the additional H&W support. To clarify this, we have added to the manuscript, including an example of when a patient may receive therapy plus H&W vs therapy only to the 'Intervention' section of the Methods (lines 126-125 and 130-136). Due to the difference in sample size between those who received therapy only versus therapy plus H&W, it would not be appropriate to statistically compare these two sub-groups as is suggested in comment 1. This is why we provided a descriptive comparison instead, as it was important to highlight.

Although the intervention group is not necessarily defined by just those receiving the H&W support, we do however understand that the relatively low percentage of patients who did receive the H&W support has implications for our findings. We have added to the limitations to highlight this (lines 408-412).

Response to Reviewer 1 comment 2

We have added to the intervention description section of the Methods, detailing the role of the PWP (lines 130-134). In addition, we have referred readers to the qualitative evaluation paper, which has been published since submitting this manuscript, which provides a more detailed overview of the referral process and components of the H&W pathway (line 143). The published paper can be found here: Perspectives on an enhanced 'Improving Access to Psychological Therapies' (IAPT) service addressing the wider determinants of mental health: a qualitative study (springer.com). We now also refer to the qualitative evaluation in the discussion (lines 314-316 and 346-350) and use the results of the qualitative evaluation to help interpret our quantitative findings.

Comments from Reviewer 2

Comment 1: 'Thank you for allowing me to review this significant manuscript entitled "Estimating the Effectiveness of an Enhanced 'Improving Access to Psychological Therapies (IAPT) Service Addressing the Wider Determinants of Mental Health: A Real-World Evaluation." This paper investigates the effectiveness of the enhanced interventions within the 'Healthy Living Healthy Minds' program and 'Wellbeing Navigator' sessions in comparison to the standard IAPT service. The study evaluates the enhanced service and standard service (i.e., IAPT Treatment-As-Usual) concerning depression, anxiety, and functioning outcomes, utilizing historical and geographical comparisons. The hypotheses tested bear substantial public and political importance due to the extensive human resources involved in the IAPT and enhanced IAPT service. While I recognize the researchers' mission and this study's primary aim, a scientific paper should offer more theoretical background and detailed information for readers to comprehend its scientific implications. My initial impression of this manuscript is the absence of specific hypotheses. Merely adding something may lead to better results, but the theoretical basis for this should be elucidated.'

Response to comment 1: We have added to the Introduction (lines 101-105 and 107-109), which we believe provides a more theoretical basis for the inclusion of health and wellbeing support to be embedded within mental health services. We have also added to the end of the Intro stating our hypothesis explicitly (lines 120-122). Essentially, poor mental health/mental health disorders could be caused or worsened by external wider determinants of people's lives, such as poverty, social isolation, debt, poor physical health, physical inactivity etc. Therefore, the hypothesis is that addressing these factors within a mental health service will help to address the root cause of the issue and create a tailored, more holistic treatment plan that is more likely to have lasting positive effects (e.g. less chance of relapse). These external factors are part of the mechanism for improving mental health and in terms of IAPT, improving and maximising patient outcomes (such as PHQ-9 and GAD-7).

Comment 2: 'Firstly, I question the selection of outcomes. The rationale for choosing depression and anxiety measures as outcomes remain unclear, as the enhanced care within the 'Healthy Living Healthy Minds' (HWHB) program and 'Wellbeing Navigator' (WBN) intervention do not seem to target depression and anxiety directly. If these are the intended outcomes, additional treatment should involve higher-intensity treatments for depression and anxiety. Is there any evidence supporting the efficacy of these interventions in improving depression and anxiety? If so, are the HWHB and WBN interventions superior to offering higher-intensity treatment (e.g., additional CBT)?'

It appears logical if the HWHB and WBN interventions target other aspects or determinants of mental health, such as well-being or quality of life, which the IAPT-TAU may not sufficiently improve. If so, the study's outcomes should focus on well-being and quality-of-life measures rather than depression and anxiety. The World Health Organisation's whole-person approach highlights the importance of examining various mental health or human living aspects. The HWHB and WBN interventions are

indeed valuable; however, their importance is more proximal to participants' daily and social health status. Depression and anxiety outcomes seem distal for these additional interventions. At the very least, these discussions prove useful when discussing this study's results, implying that different outcome evaluations are necessary to assess the true outcomes of these enhanced services.'

Response to comment 2: As we addressed in our response above, the H&W pathway is not meant to directly treat anxiety or depression but instead address the wider determinants that may be causing the mental health issues and therefore are an important part of the mechanism to improving mental health. IAPT has a key set of performance indicators and metrics, which the service is judged against (which we define in the Intro (lines 87-96)). Although PHQ-9 and GAD-7 are symptom-based measures of depression and anxiety and there perhaps should be a movement towards recovery-focused and wellbeing measures, it is PHQ-9 and GAD-7 which are the pertinent outcomes to IAPT services and this is why our evaluation using IAPT's electronic health record data focused on these as primary outcomes.

We do also include WSAS score as a primary outcome, which is a measure of social functioning/impairment and includes items relating to the ability to work, do household management, take part in leisure activities and maintain close relationships, which may relate to some of the external factors being addressed through the H&W pathway. We do agree that measures of, for example wellbeing and quality of life would have been useful for this evaluation and would have enhanced our interpretation, however these measures are not currently collected by IAPT services because the service is not judged against these outcomes. Because this was a service-led evaluation, we as a research team were unable to collect any primary data and instead used IAPT electronic health record (EHR) data to conduct our analyses, which did not include any measures such as wellbeing which we could link to the H&W support. We have added to the manuscript (line 145) clarifying that this study was a service-led public health evaluation. We state our use of the EHR data in the Methods section (line 166) and also discuss the advantages and disadvantages of using EHR data in the Limitations section (lines 400-404). We have added to the Limitations section, discussing how additional measures such as wellbeing could have helped to better understand the more mechanistic impact of the H&W pathway (lines 404-407). We have also added to the Clinical Implications section highlighting that it would be beneficial for mental health services to assess other measures such as wellbeing, in addition to anxiety and depression measures to help better understand the pathway to improved mental health (lines 427-429).

Comment 3: 'The authors should estimate the effective strength of the enhanced service based on previous studies. Without this information, readers cannot evaluate whether the study has an adequate sample size or whether the benefits observed are "small" or not.'

Response to comment 3: The ATTs observed in our study were small, which we state in the discussion (line 337). The results represented a less than 1 point score difference in mental health improvement between groups. There are limited previous studies that evaluate the effectiveness of adding other support to address wider determinants of mental health alongside therapy on clinical mental health outcomes such as PHQ-9, GAD-7 and WSAS. It would not be appropriate for us to directly compare estimates with other studies that use different measures of mental health nor studies that compare different treatment options (e.g. counselling vs CBT). However, we can make some general observations around direction and size of effect, and we have added to the discussion comparing our estimates to one similar study that explored adding insomnia treatment alongside therapy (lines 337-341).

Comment 4: 'Details on the interventions are scant. I could not ascertain the nature of the six-session group webinar series within the 'Healthy Living Healthy Minds' program and one-to-one sessions of 'Wellbeing Navigator.' What were the contents? Who provided these services? How were these

providers trained? What underpinning theories were used for these interventions? How was fidelity assessed for these interventions? How long did each session last? What types of psychosocial problems were addressed in the intervention group? What was the frequency of sessions? How long did the intervention last? Was there any interaction between participants in the 'Healthy Living Healthy Minds' program?'

Response to comment 4: We have addressed this comment in a response to Reviewer 1. Please see the 'Response to Reviewer 1 comment 2' above.

5) 'I acknowledge that reliable change and recovery have been used as IAPT outcomes. However, I require clarification on the differences between these two outcomes. Why were both outcomes necessary? In the discussion section, the distinction between mean PHQ-9 and GAD-7 scores and recovery status was helpful. I understand that the mean score represents the average improvement observed at the group level, while reliable change and recovery indicate the proportion of participants who responded to the intervention. Reliable change pertains to participants who responded to the intervention, whereas recovery pertains to participants who scored below the cut-offs. Is this understanding correct? If so, the proportion of recovery should always be lower than the reliable change. However, the results do not consistently support this in every group, possibly due to using both PHQ-9 and GAD-7 scores to define recovery status. The actual value of the proportion is not significantly different, leading me to question the importance of employing these two outcomes. Are there any specific hypotheses based on these different outcomes?'

Response to comment 5: Reliable change and moving from caseness are two separate and key performance indicators of the IAPT service (lines 87-96). To clarify the definitions, reliable change requires a score reduction of 6 or more, whereas recovery is defined as moving from 'caseness' to 'no caseness', with caseness based on IAPT-defined cut-offs. Therefore, it is not necessarily the case that proportion of patients achieving recovery should always be lower than the proportion of patients achieving reliable change. For example, a patient with a baseline score of 11 on PHQ-9 (classified as caseness) could reduce their score by only 2 points over the course of their treatment and would then be defined by the IAPT service as 'recovered' because their follow-up score is below the cut-off point for caseness (no caseness = score of less than 10), however they would not have achieved reliable change because this requires a score reduction of 6 or more.

Reliable change and recovery are two key outcomes that are assessed within IAPT services and for which IAPT service providers will be held accountable to. Reliable change and recovery are two of the key statistics reported in NHS England's annual IAPT reports, as there is a national target for all IAPT services to achieve a minimum of 50% recovery rate (line 97). Therefore, as this was a service-led public health evaluation, reliable change and recovery were deemed key outcomes for this study and we followed the NHS England's IAPT Manual to calculate these outcomes. We clearly define reliable change and recovery in the Introduction and state that these outcomes are part of IAPT's key performance indicators (lines 87-96). We also refer to the IAPT manual in the Measures section of the Methods when describing reliable change and recovery as secondary outcomes. We have edited this section to emphasise these are two separate key performance indicators (lines 175-178).

6) 'Please clarify why reliable change and recovery status was calculated only for those who reached the point of service discharge. This approach appears to be a completer analysis. It may be more appropriate to use an intention-to-treat (ITT) analysis incorporating all participants who started the service for statistical analyses.'

Response to comment 6: As mentioned above, we state in the Measures section of Methods that these outcomes were calculated for those who had been discharged as per the IAPT Manual. As this was a service-led evaluation, we followed guidance from the IAPT Manual and collaborators who

were involved in transferring the IAPT electronic health record data, whose expertise we draw from when cleaning the data and calculating variables. We have reworded this section to make clearer that this was because we were following IAPT guidance on how to calculate outcomes (lines 175-178).

To note, as we state in the Statistical analysis section of the Methods (lines 212-215), all our analyses are based on a complete-case analysis rather than an intention-to-treat analysis. We deemed a complete-case analysis to be preferential over an intention-to-treat analysis because 1) To align with the IAPT Manual and IAPT's KPI's for recovery, 2) Because this study is a real-world evaluation using electronic health record data, where patients' baseline and follow-up data could fall anywhere within our timeframe of study. It was therefore important to specify a follow up timepoint for our analysis. We took a practical, service-led approach in defining follow-up as point of service discharge in the primary analysis and then also used 12, 16, 20 and 24 weeks as follow ups in the sensitivity analyses. If a follow-up time point were not to be used (i.e. a ITT), patient follow-up times could vary greatly and for example include patients who entered the service right before the end of our timeframe of study, which would have biased our estimates. We've empathised this to the Statistical Analysis section (line 215) and we acknowledge and discuss the limitations of this approach in the Limitations section (lines 387-399).

7) 'I require an explanation of why waiting times could be considered an outcome. Please provide the rationale for hypothesizing that enhanced care is superior to TAU in this outcome and include an explanation.'

Response to comment 7: Similar to our response to comment 5, waiting time is a key performance indicator for IAPT and so was important to include in this service-led public health evaluation. We hypothesised that waiting times would be reduced in the enhanced service because patients could be offered a first appointment sooner, which was part of the H&W pathway whilst waiting for their therapy to start. Waiting time is defined as time until first appointment and an appointment as part of the H&W pathway would be considered the first appointment (lines 178-179). We have added this to our hypothesis statement at the end of the Intro (lines 120-122).

Comment 8: *'Please include explanations of NB and IMD for Table 1.'*

Response to comment 8: We have added these to the footnote under Table 1.

9) 'In the introduction section, the authors mention that approximately 50% of the referred 1.81 million adults were defined as 'recovered' at the point of discharge, while the historical control group in this study achieved only 38% 'recovered' status. This discrepancy suggests that the enhanced service's superior results may be due to unusually weak results in the historical group. The authors should address this possibility. If this is likely, the significant results do not reflect the effectiveness of the enhanced service; rather, they may indicate the historical group's low-quality IAPT service provision. Although I understand that the researchers lack data on the quality and adherence of IAPT services specific to this study, are there any other findings regarding IAPT service quality that could help interpret the historical and geographical aspects?'

Response to comment 9: In the discussion, we explain that the group difference between intervention and historical control is likely to be due to more general improvements to the service over time (as a new service provider took over) rather than due to the addition of the H&W pathway specifically (see lines 323-330). We also include this in our conclusion (line 431-432). We have added to the discussion noting the low recovery rates in the historical control service sample to clarify this point (lines 331-332).

VERSION 2 – REVIEW

REVIEWER	Eiji Shimizu Chiba University Graduate School of Medicine School of Medicine, Department of Cognitive Behavioral Physiology, Graduate School of Medicine
REVIEW RETURNED	22-Aug-2023
GENERAL COMMENTS	The authors have adequately responded all of the reviewer's comments.